# Learnable Topological Features for Phylogenetic Inference via Graph Neural Networks

**Cheng Zhang**
School of Mathematical Sciences and Center for Statistical Science
Peking University, Beijing, China
`chengzhang@math.pku.edu.cn`

## Abstract

Structural information of phylogenetic tree topologies plays an important role in phylogenetic inference. However, finding appropriate topological structures for specific phylogenetic inference tasks often requires significant design effort and domain expertise. In this paper, we propose a novel structural representation method for phylogenetic inference based on learnable topological features. By combining the raw node features that minimize the Dirichlet energy with modern graph representation learning techniques, our learnable topological features can provide efficient structural information of phylogenetic trees that automatically adapts to different downstream tasks without requiring domain expertise. We demonstrate the effectiveness and efficiency of our method on a simulated data tree probability estimation task and a benchmark of challenging real data variational Bayesian phylogenetic inference problems.

## 1 Introduction

Phylogenetics is an important discipline of computational biology where the goal is to identify the evolutionary history and relationships among individuals or groups of biological entities. In statistical approaches to phylogenetics, this has been formulated as an inference problem on hypotheses of shared history, i.e., *phylogenetic trees*, based on observed sequence data (e.g., DNA, RNA, or protein sequences) under a model of evolution. The phylogenetic tree defines a probabilistic graphical model, based on which the likelihood of the observed sequences can be efficiently computed (Felsenstein, 2003). Many statistical inference procedures therefore can be applied, including maximum likelihood and Bayesian approaches (Felsenstein, 1981; Yang & Rannala, 1997; Mau et al., 1999; Huelsenbeck et al., 2001).

Phylogenetic inference, however, has been challenging due to the composite parameter space of both continuous and discrete components (i.e., branch lengths and the tree topology) and the combinatorial explosion in the number of tree topologies with the number of sequences. Harnessing the topological information of trees hence becomes crucial in the development of efficient phylogenetic inference algorithms. For example, by assuming conditional independence of separated subtrees, Larget (2013) showed that conditional clade distributions (CCDs) can provide more reliable tree probability estimation that generalizes beyond observed samples. A similar approach was proposed to design more efficient proposals for tree movement when implementing Markov chain Monte Carlo (MCMC) algorithms for Bayesian phylogenetics (Höhna & Drummond, 2012). Utilizing more sophisticated local topological structures, CCDs were later generalized to subsplit Bayesian networks (SBNs) that provide more flexible distributions over tree topologies (Zhang & Matsen IV, 2018). Besides MCMC, variational Bayesian phylogenetics inference (VBPI) was recently proposed that leveraged SBNs and a structured amortization of branch lengths to deliver competitive posterior estimates in a more timely manner (Zhang & Matsen IV, 2019; Zhang, 2020; Zhang & Matsen IV, 2022). Azouri et al. (2021) used a machine learning approach to accelerate maximum likelihood tree-search algorithms by providing more informative topology moves. Topological features have also been found useful for comparison and interpretation of the reconstructed phylogenies (Matsen IV, 2007; Hayati et al., 2022). While these approaches prove effective in practice, they all rely on heuristic features (e.g., clades and subsplits) of phylogenetic trees that often require significant design effort and domain expertise, and may be insufficient for capturing complicated topological information.

Graph Neural Networks (GNNs) are an effective framework for learning representations of graph-structured data. To encode the structural information about graphs, GNNs follow a neighborhood aggregation procedure that computes the representation vector of a node by recursively aggregating and transforming representation vectors of its neighboring nodes. After the final iteration of aggregation, the representation of the entire graph can also be obtained by pooling all the node embeddings together via some permutation invariant operators (Ying et al., 2018). Many GNN variants have been proposed and have achieved superior performance on both node-level and graph-level representation learning tasks (Kipf & Welling, 2017; Hamilton et al., 2017; Li et al., 2016; Zhang et al., 2018; Ying et al., 2018). A natural idea, therefore, is to adapt GNNs to phylogenetic models for automatic topological feature learning. However, the lack of node features for phylogenetic trees makes it challenging as most GNN variants assume fully observed node features at initialization.

In this paper, we propose a novel structural representation method for phylogenetic inference that automatically learns efficient topological features based on GNNs. To obtain the initial node features for phylogenetic trees, we follow previous studies (Zhu & Ghahramani, 2002; Rossi et al., 2021) to minimize the Dirichlet energy, with one hot encoding for the tip nodes. Unlike these previous studies, we present a fast linear time algorithm for Dirichlet energy minimization by taking advantage of the hierarchical structure of phylogenetic trees. Moreover, we prove that these features are sufficient for identifying the corresponding tree topology, i.e., there is no information loss in our raw feature representations of phylogenetic trees. These raw node features are then passed to GNNs for more sophisticated structure representation learning required by downstream tasks. Experiments on a synthetic data tree probability estimation problem and a benchmark of challenging real data variational Bayesian phylogenetic inference problems demonstrate the effectiveness and efficiency of our method.

## 2 BACKGROUND

**Notation**   A phylogenetic tree is denoted as $(\tau, \boldsymbol{q})$ where $\tau$ is a bifurcating tree that represents the evolutionary relationship of the species and $\boldsymbol{q}$ is a non-negative branch length vector that characterizes the amount of evolution along the edges of $\tau$. The tip nodes of $\tau$ correspond to the observed species and the internal nodes of $\tau$ represent the unobserved characters (e.g., DNA bases) of the ancestral species. The transition probability $P_{ij}(t)$ from character $i$ to character $j$ along an edge of length $t$ is often defined by a continuous-time substitution model (e.g., Jukes & Cantor (1969)), whose stationary distribution is denoted as $\eta$. Let $E(\tau)$ be the set of edges of $\tau$, $r$ be the root node (or any internal node if the tree is unrooted and the substitution model is reversible). Let $\boldsymbol{Y} = \{Y_1, Y_2, \ldots, Y_M\} \in \Omega^{N \times M}$ be the observed sequences (with characters in $\Omega$) of length $M$ over $N$ species.

**Phylogenetic posterior**   Assuming different sites $Y_i, i = 1, \ldots, M$ are independent and identically distributed, the likelihood of observing $\boldsymbol{Y}$ given the phylogenetic tree $(\tau, \boldsymbol{q})$ takes the form

$$p(\boldsymbol{Y}|\tau, \boldsymbol{q}) = \prod_{i=1}^{M} p(Y_i|\tau, \boldsymbol{q}) = \prod_{i=1}^{M} \sum_{a^i} \eta(a_r^i) \prod_{(u,v) \in E(\tau)} P_{a_u^i a_v^i}(q_{uv}), \tag{1}$$

where $a^i$ ranges over all extensions of $Y_i$ to the internal nodes with $a_u^i$ being the assigned character of node $u$. The above phylogenetic likelihood function can be computed efficiently through the pruning algorithm (Felsenstein, 2003). Given a prior distribution $p(\tau, \boldsymbol{q})$ of the tree topology and the branch lengths, Bayesian phylogenetics then amounts to properly estimating the phylogenetic posterior $p(\tau, \boldsymbol{q}|\boldsymbol{Y}) \propto p(\boldsymbol{Y}|\tau, \boldsymbol{q})p(\tau, \boldsymbol{q})$.

**Variational Bayesian phylogenetic inference**   Let $Q_{\boldsymbol{\phi}}(\tau)$ be an SBN-based distribution over the tree topologies and $Q_{\boldsymbol{\psi}}(\boldsymbol{q}|\tau)$ be a non-negative distribution over the branch lengths. VBPI finds the best approximation to $p(\tau, \boldsymbol{q}|\boldsymbol{Y})$ from the family of products of $Q_{\boldsymbol{\phi}}(\tau)$ and $Q_{\boldsymbol{\psi}}(\boldsymbol{q}|\tau)$ by maximizing the following multi-sample lower bound

$$L^K(\boldsymbol{\phi}, \boldsymbol{\psi}) = \mathbb{E}_{Q_{\boldsymbol{\phi}, \boldsymbol{\psi}}(\tau^{1:K}, \boldsymbol{q}^{1:K})} \log \left( \frac{1}{K} \sum_{i=1}^{K} \frac{p(\boldsymbol{Y}|\tau^i, \boldsymbol{q}^i) p(\tau^i, \boldsymbol{q}^i)}{Q_{\boldsymbol{\phi}}(\tau^i) Q_{\boldsymbol{\psi}}(\boldsymbol{q}^i|\tau^i)} \right) \leq \log p(\boldsymbol{Y}) \tag{2}$$

where $Q_{\boldsymbol{\phi}, \boldsymbol{\psi}}(\tau^{1:K}, \boldsymbol{q}^{1:K}) = \prod_{i=1}^{K} Q_{\boldsymbol{\phi}}(\tau^i) Q_{\boldsymbol{\psi}}(\boldsymbol{q}^i|\tau^i)$. To properly parameterize the variational distributions, a support of the conditional probability tables (CPTs) is often acquired from a sample

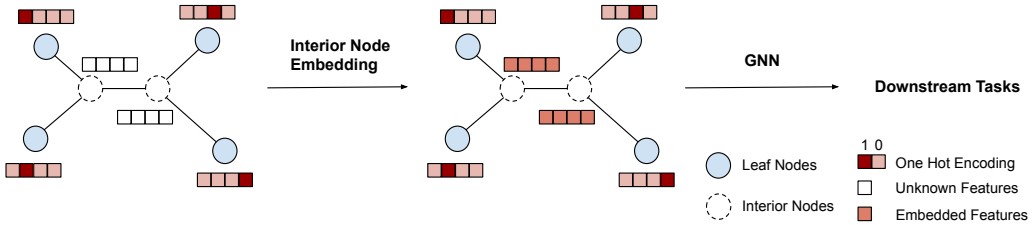

Figure 1: An overview of the proposed topological feature learning framework for phylogenetic inference. **Left**: A phylogenetic tree topology with one hot encoding for the tip nodes and missing features for the interior nodes. **Middle**: Interior node embedding via Dirichlet energy minimization. **Right**: Subsequently, the tree topology with embedded node features are fed into a GNN model for more sophisticated tree structure representation learning required by downstream tasks.

of tree topologies via fast heuristic bootstrap methods (Minh et al., 2013; Zhang & Matsen IV, 2019). The branch length approximation $Q_{\boldsymbol{\psi}}(\boldsymbol{q}|\tau)$ is taken to be the diagonal Lognormal distribution

$$Q_{\boldsymbol{\psi}}(\boldsymbol{q}|\tau) = \prod_{e \in E(\tau)} p^{\text{Lognormal}}\left(q_e \mid \mu(e, \tau), \sigma(e, \tau)\right)$$

where $\mu(e, \tau), \sigma(e, \tau)$ are amortized over the tree topology space via shared local structures (i.e., split and primary subsplit pairs (PSPs)), which are available from the support of CPTs. More details about structured amortization, VBPI and SBNs can be found in section 3.2.2 and Appendix A.

**Graph neural networks** Let $G = (V, E)$ denote a graph with node feature vectors $\boldsymbol{X}_v$ for node $v \in V$, and $\mathcal{N}(v)$ denote the set of nodes adjacent to $v$. GNNs iteratively update the representation of a node by running a message passing (MP) scheme for $T$ time steps. During each MP time step, the representation vectors of each node are updated based on the aggregated messages from its neighbors as follows

$$\boldsymbol{h}_v^{(t+1)} = \text{UPDATE}^{(t)}\left(\boldsymbol{h}_v^{(t)}, \boldsymbol{m}_v^{(t+1)}\right), \quad \boldsymbol{m}_v^{(t+1)} = \text{AGG}^{(t)}\left(\left\{\boldsymbol{h}_u^{(t)} : u \in \mathcal{N}(v)\right\}\right)$$

where $\boldsymbol{h}_v^{(t)}$ is the feature vector of node $v$ at time step $t$, with initialization $\boldsymbol{h}_v^{(0)} = \boldsymbol{X}_v$, UPDATE$^{(t)}$ is the update function, and AGG$^{(t)}$ is the aggregation function. A number of powerful GNNs with different implementations of the update and aggregation functions have been proposed (Kipf & Welling, 2017; Hamilton et al., 2017; Li et al., 2016; Veličković et al., 2018; Xu et al., 2019; Wang et al., 2019). In additional to the local node-level features, GNNs can also provide features for the entire graph. To learn these global features, an additional READOUT function is often introduced to aggregate node features from the final iteration

$$\boldsymbol{h}_G = \text{READOUT}\left(\left\{\boldsymbol{h}_v^{(T)} : v \in V\right\}\right).$$

READOUT can be any function that is permutation invariant to the node features.

## 3 PROPOSED METHOD

In this section, we propose a general approach that automatically learns topological features directly from phylogenetic trees. We first introduce a simple embedding method that provides raw features for the nodes of phylogenetic trees, together with an efficient linear time algorithm for obtaining these raw features and a discussion on some of their theoretical properties regarding tree topology representation. We then describe how these raw features can be adapted to learn efficient representations of certain structures of trees (e.g., edges) for downstream tasks.

### 3.1 INTERIOR NODE EMBEDDING

Learning tree structure features directly from tree topologies often requires raw node/edge features, as typically assumed in most GNN models. Unfortunately, this is not the case for phylogenetic models.

Although we can use one hot encoding for the tip nodes according to their corresponding species (taxa names only, not the sequences), the interior nodes still lack original features. The first step of tree structure representation learning for phylogenetic models, therefore, is to properly input those missing features for the interior nodes. Following previous studies (Zhu & Ghahramani, 2002; Rossi et al., 2021), we make a common assumption that the node features change smoothly across the tree topologies (i.e., the features of every node are similar to those of the neighbors). A widely used criterion of smoothness for functions defined on nodes of a graph is the *Dirichlet energy*. Given a tree topology $\tau = (V, E)$ and a function $f : V \mapsto \mathbb{R}^d$, the Dirichlet energy is defined as

$$\ell(f, \tau) = \sum_{(u,v) \in E} \|f(u) - f(v)\|^2.$$

Let $V = V^b \cup V^o$, where $V^b$ denotes the set of leaf nodes and $V^o$ denotes the set of interior nodes. Let $\boldsymbol{X}^b = \{\boldsymbol{x}_v | v \in V^b\}$ be the set of one hot embeddings for the leaf nodes. The interior node features $\boldsymbol{X}^o = \{\boldsymbol{x}_v | v \in V^o\}$ then can be obtained by minimizing the Dirichlet energy

$$\widehat{\boldsymbol{X}^o} = \arg\min_{\boldsymbol{X}^o} \ell(\boldsymbol{X}^o, \boldsymbol{X}^b, \tau) = \arg\min_{\boldsymbol{X}^o} \sum_{(u,v) \in E} \|\boldsymbol{x}_u - \boldsymbol{x}_v\|^2.$$

### 3.1.1 A Linear Time Two-pass Algorithm

Note that the above Dirichlet energy function is convex, its minimizer therefore can be obtained by solving the following optimality condition

$$\frac{\partial \ell(\boldsymbol{X}^o, \boldsymbol{X}^b, \tau)}{\partial \boldsymbol{X}^o}(\widehat{\boldsymbol{X}^o}) = \boldsymbol{0}. \tag{3}$$

It turns out that equation 3 has a close-form solution based on matrix inversion. However, as matrix inversion scales cubically in general, it is infeasible for graphs with many nodes. Fortunately, by leveraging the hierarchical structure of phylogenetic trees, we can design a more efficient linear time algorithm for the solution of equation 3 as follows. We first rewrite equation 3 as a system of linear equations

$$\sum_{v \in \mathcal{N}(u)} (\widehat{\boldsymbol{x}}_u - \widehat{\boldsymbol{x}}_v) = \boldsymbol{0}, \quad \forall u \in V^o, \qquad \widehat{\boldsymbol{x}}_v = \boldsymbol{x}_v, \quad \forall v \in V^b, \tag{4}$$

where $\mathcal{N}(u)$ is the set of neighbors of node $u$. Given a topological ordering induced by the tree[1], we can obtain the solution within a two-pass sweep through the tree topology, similar to the Thomas algorithm for solving tridiagonal systems of linear equations (Thomas, 1949). In the first pass, we traverse the tree in a postorder fashion and express the node features as a linear function of those of their parents,

$$\widehat{\boldsymbol{x}}_u = c_u \widehat{\boldsymbol{x}}_{\pi_u} + \boldsymbol{d}_u, \tag{5}$$

for all the nodes expect the root node, where $\pi_u$ denotes the parent node of $u$. More specifically, we first initialize $c_u = 0, \boldsymbol{d}_u = \boldsymbol{x}_u$ for all leaf nodes $u \in V^b$. For all the interior nodes except the root node, we compute $c_u, \boldsymbol{d}_u$ recursively as follows (see a detailed derivation in Appendix B)

$$c_u = \frac{1}{|\mathcal{N}(u)| - \sum_{v \in \text{ch}(u)} c_v}, \quad \boldsymbol{d}_u = \frac{\sum_{v \in \text{ch}(u)} \boldsymbol{d}_v}{|\mathcal{N}(u)| - \sum_{v \in \text{ch}(u)} c_v}, \tag{6}$$

where $\text{ch}(u)$ denotes the set of child nodes of $u$. In the second pass, we traverse the tree in a preorder fashion and compute the solution by back substitution. Concretely, at the root node $r$, given equation 5 for all the child nodes from the first pass, we can compute the node feature directly from equation 4 as below

$$\widehat{\boldsymbol{x}}_r = \frac{\sum_{v \in \text{ch}(r)} \boldsymbol{d}_v}{|\mathcal{N}(r)| - \sum_{v \in \text{ch}(r)} c_v}. \tag{7}$$

For all the other interior nodes, the node features can be obtained via equation 5 by substituting the learned features for the parent nodes. We summarize our two-pass algorithm in Algorithm 1. Moreover, the algorithm is numerically stable due to the following lemma (proof in Appendix C).

**Lemma 1.** *Let* $\lambda = \min_{u \in V^o \setminus \{r\}} |\mathcal{N}(u)|$. *For all interior node* $u \in V^o \setminus \{r\}$, $0 \leq c_u \leq \frac{1}{\lambda - 1}$.

Besides bifurcating phylogenetic trees, the above two-pass algorithm can be easily adapted to interior node embedding for general tree-shaped graphs with given tip node features.

---

[1]This is trivial for rooted trees since they are directed. For unrooted trees, we can choose an interior node as the root node and use the topological ordering of the corresponding rooted trees.

---

**Algorithm 1** A Two-pass Algorithm for Interior Node Embedding

1: **Input:** Tree topology $\tau = (V, E)$, where $V = V^b \cup V^o$; Features for the tip nodes $\{\boldsymbol{x}_u | u \in V^b\}$.
2: Initialize $c_u = 0, \boldsymbol{d}_u = \boldsymbol{x}_u, \forall u \in V^b$.
3: Traverse the tree topology in a postorder fashion. For any interior node $u$ that is not the root node, compute $c_u$ and $\boldsymbol{d}_u$ as in equation 6.
4: Traverse the tree topology in a preorder fashion. For the root node $r$, compute the node feature as in equation 7. For any other interior node $u$, compute the node feature as $\widehat{\boldsymbol{x}}_u = c_u \widehat{\boldsymbol{x}}_{\pi_u} + \boldsymbol{d}_u$.
5: **return** $\{\widehat{\boldsymbol{x}}_u | u \in V^o\}$.

---

### 3.1.2 TREE TOPOLOGY REPRESENTATION POWER

In this section, we discuss some theoretical properties regarding the tree topology presentation power of the node features introduced above. We start with a useful lemma that elucidates an important behavior of the solution to the linear system 4, which is similar to the solutions to elliptic equations.

**Lemma 2** (Extremum Principle). *Let $\{\widehat{\boldsymbol{x}}_u \in \mathbb{R}^d | u \in V\}$ be a set of $d$-dimensional node features that satisfies equations 4. $\forall 1 \leq n \leq d$, let $\widehat{\boldsymbol{X}}[n] = \{\widehat{\boldsymbol{x}}_u[n] | u \in V\}$ be the set of the $n$-th components of node features. Then, $\forall 1 \leq n \leq d$, we have: (i) the extremum values (i.e., maximum and minimum) of $\widehat{\boldsymbol{X}}[n]$ can be achieved at some tip nodes; (ii) if the extremum values are achieved at some interior nodes, then $\widehat{\boldsymbol{X}}[n]$ has only one member, or in other words, $\widehat{\boldsymbol{x}}_u[n]$ is the same $\forall u \in V$.*

**Theorem 1.** *Let $N$ be the number of tip nodes. Let $\{\widehat{\boldsymbol{x}}_u \in \mathbb{R}^N | u \in V\}$ be the solution to the linear system 4 with one hot encoding for the tip nodes. Then, $\forall u \in V^o$, we have*

$$(i) \ 0 < \widehat{\boldsymbol{x}}_u[n] < 1, \quad \forall 1 \leq n \leq N, \quad and \quad (ii) \sum_{n=1}^{N} \widehat{\boldsymbol{x}}_u[n] = 1.$$

The complete proofs of Lemma 2 and Theorem 1 are provided in Appendix C. When the tip node features are linearly independent, a similar proposition holds when we consider the coefficients of the linear combination of the tip node features for the interior node features instead.

**Corollary 1.** *Suppose that the tip node features are linearly independent, the interior node features obtained from the solution to the linear system 4 all lie in the interior of the convex hull of all tip node features.*

The proof is provided in Appendix C. The following lemma reveals a key property of the nodes that are adjacent to the boundary of the tree topology in the embedded feature space.

**Lemma 3.** *Let $\{\widehat{\boldsymbol{x}}_u | u \in V\}$ be the solution to the linear system 4, with linearly independent tip node features. Let $\{\widehat{\boldsymbol{x}}_u = \sum_{v \in V^b} a_u^v \boldsymbol{x}_v | u \in V^o\}$ be the convex combination representations of the interior node features. For any tip node $v \in V^b$, we have*

$$u^* = \arg\max_{u \in V^o} a_u^v \quad \Leftrightarrow \quad u^* \in \mathcal{N}(v).$$

**Theorem 2** (Identifiability). *Let $\boldsymbol{X}^o = \{\widehat{\boldsymbol{x}}_u | u \in V^o\}$ and $\boldsymbol{Z}^o = \{\widehat{\boldsymbol{z}}_u | u \in V^o\}$ be the sets of interior node features that minimizes the Dirichlet energy for phylogenetic tree topologies $\tau_x$ and $\tau_z$ respectively, given the same linearly independent tip node features. If $\boldsymbol{X}^o = \boldsymbol{Z}^o$, then $\tau_x = \tau_z$.*

The proofs of Lemma 3 and Theorem 2 are provided in Appendix C. By Theorem 2, we see that the proposed node embeddings are complete representations of phylogenetic tree topologies with no information loss.

### 3.2 STRUCTURAL REPRESENTATION LEARNING VIA GRAPH NEURAL NETWORKS

Using node embeddings introduced in section 3.1 as raw features, we now show how to learn more sophisticated representations of tree structures for different phylogenetic inference tasks via GNNs. Given a tree topology $\tau$, let $\{\boldsymbol{h}_v^{(0)} : v \in V\}$ be the raw features and $\{\boldsymbol{h}_v^{(T)} : v \in V\}$ be the output features after the final iteration of GNNs. We feed these output features of GNNs into a multi-layer perceptron (MLP) to get a set of learnable features for each node

$$\boldsymbol{h}_v = \text{MLP}^{(0)}\left(\boldsymbol{h}_v^{(T)}\right), \quad \forall v \in V,$$

before adapting to different downstream tasks, as demonstrated in the following examples.

### 3.2.1 ENERGY BASED MODELS FOR TREE PROBABILITY ESTIMATION

Our first example is on graph-level representation learning of phylogenetic tree topologies. Let $\mathcal{T}$ denote the entire tree topology space. Given learnable node features of tree topologies, one can use a permutation invariant function $g$ to obtain graph-level features and hence create an energy function $F_{\phi} : \mathcal{T} \mapsto \mathbb{R}$ that assigns each tree topology a scalar value as follows

$$F_{\phi}(\tau) = \text{MLP}^{(1)}(\boldsymbol{h}_G), \quad \boldsymbol{h}_G = g\left(\{\boldsymbol{h}_v : v \in V\}\right).$$

where $g \circ \text{MLP}^{(0)}$ can be viewed as a READOUT function in section 2. This allows us to construct energy based models (EBMs) for tree probability estimation

$$q_{\phi}(\tau) = \frac{\exp\left(-F_{\phi}(\tau)\right)}{Z(\phi)}, \quad Z(\phi) = \sum\nolimits_{\tau \in \mathcal{T}} \exp\left(-F_{\phi}(\tau)\right).$$

As $Z(\phi)$ is usually intractable, we can employ noise contrastive estimation (NCE) (Gutmann & Hyvärinen, 2010) to train these energy based models. Let $p_n$ be some noise distribution that has tractable density and allows efficient sampling procedures. Let $D_{\phi}(\tau) = \log q_{\phi}(\tau) - \log p_n(\tau)$. We can train $D_{\phi}$ [2] to minimize the following objective function (NCE loss)

$$J(\phi) = -\left(\mathbb{E}_{\tau \sim p_{\text{data}}(\tau)} \log\left(S\left(D_{\phi}(\tau)\right)\right) + \mathbb{E}_{\tau \sim p_n(\tau)} \log\left(1 - S\left(D_{\phi}(\tau)\right)\right)\right),$$

where $S(x) = \frac{1}{1+\exp(-x)}$ is the sigmoid function. It is easy to verify that the minimum is achieved at $D_{\phi^*}(\tau) = \log p_{\text{data}}(\tau) - \log p_n(\tau)$. Therefore, $q_{\phi^*}(\tau) = p_{\text{data}}(\tau) = p_n(\tau) \exp\left(D_{\phi^*}(\tau)\right)$.

### 3.2.2 BRANCH LENGTH PARAMETERIZATION FOR VBPI

The branch length parameterization in VBPI so far has relied on hand-engineered features (i.e., splits and PSPs) for the edges on tree topologies. Let $\mathbb{S}_r$ denote the set of splits and $\mathbb{S}_{\text{psp}}$ denote the set of PSPs. The simple split-based parameterization assigns parameters $\boldsymbol{\psi}^{\mu}, \boldsymbol{\psi}^{\sigma}$ for splits in $\mathbb{S}_r$. The mean and standard deviation for each edge $e$ on $\tau$ are then given by the associated parameters of the corresponding split $e/\tau$ as follows

$$\mu(e, \tau) = \psi_{e/\tau}^{\mu}, \quad \sigma(e, \tau) = \psi_{e/\tau}^{\sigma}. \tag{8}$$

The more flexible PSP parameterization assigns additional parameters for PSPs in $\mathbb{S}_{\text{psp}}$ and adds the associated parameters of the corresponding PSPs $e /\!/ \tau$ to equation 8 to refine the mean and standard deviation parameterization

$$\mu(e, \tau) = \psi_{e/\tau}^{\mu} + \sum\nolimits_{s \in e /\!/ \tau} \psi_s^{\mu}, \quad \sigma(e, \tau) = \psi_{e/\tau}^{\sigma} + \sum\nolimits_{s \in e /\!/ \tau} \psi_s^{\sigma}. \tag{9}$$

Although these heuristic features prove effective, they often require substantial design effort, a sample of tree topologies for feature collection, and can not adapt themselves during training which makes it difficult for amortized inference over different tree topologies. Based on the learnable node features, we can design a more flexible branch length parameterization that is capable of distilling more effective structural information of tree topologies for variational approximations. For each edge $e = (u, v)$ on $\tau$, similarly as in section 3.2.1, one can use a permutation invariant function $f$ to obtain edge-level features and transform them into the mean and standard deviation parameters as follows

$$\mu(e, \tau) = \text{MLP}^{\mu}\left(\boldsymbol{h}_e\right), \quad \sigma(e, \tau) = \text{MLP}^{\sigma}\left(\boldsymbol{h}_e\right), \quad \boldsymbol{h}_e = f\left(\{\boldsymbol{h}_u, \boldsymbol{h}_v\}\right). \tag{10}$$

Compared to heuristic feature based parameterizations in 8 and 9, learnable topological feature based parameterizations in 10 allow much richer design for the branch length distributions across different tree topologies and do not require pre-sampled tree topologies for feature collection.

## 4 EXPERIMENTS

In this section, we test the effectiveness and efficiency of learnable topological features for phylogenetic inference on the two aforementioned benchmark tasks: tree probability estimation via energy based models and branch length parameterization for VBPI. Following Zhang & Matsen IV (2019), in VBPI we used the simplest SBN for the tree topology variational distribution, and the CPT supports

---

[2]Here $Z(\phi)$ is taken as a free parameter and is included into $\phi$.

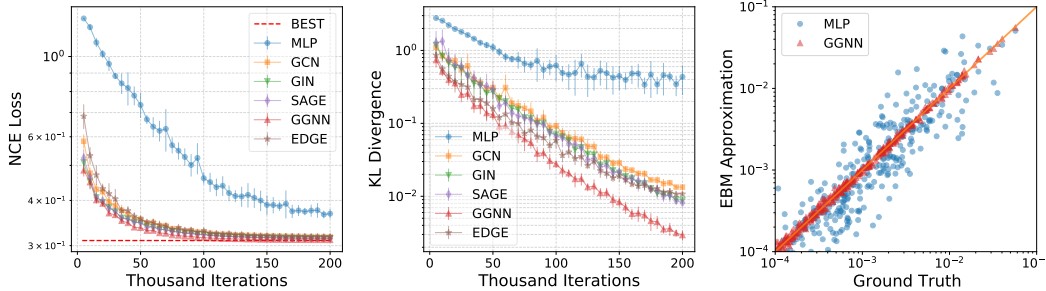

Figure 2: Comparison of learnable topological feature based EBMs for probability mass estimation of unrooted phylogenetic trees with 8 leaves using NCE. **Left:** NCE loss. **Middle:** KL divergence. **Right:** EBM approximations vs ground truth probabilities. The NCE loss and KL divergence results were obtained from 10 independent runs and the error bars represent one standard deviation.

were estimated from ultrafast maximum likelihood phylogenetic bootstrap trees using UFBoot (Minh et al., 2013). The code is available at `https://github.com/zcrabbit/vbpi-gnn`.

**Experimental setup.**   We evaluate five commonly used GNN variants with the following convolution operators: graph convolution networks (GCN), graph isomorphism operator (GIN), GraphSAGE operator (SAGE), gated graph convolution operator (GGNN) and edge convolution operator (EDGE). See more details about these convolution operators in Appendix F. In addition to the above GNN variants, we also considered a simpler model that skips all GNN iterations (i.e., $T = 0$) and referred to it as MLP in the sequel. All GNN variants have 2 GNN layers (including the input layer), and all involved MLPs have 2 layers. We used summation as our permutation invariant aggregation function for graph-level features and maximization for edge-level features. All models were implemented in Pytorch (Paszke et al., 2019) with the Adam optimizer (Kingma & Ba, 2015).We designed our experiments with the goals of (i) verifying the effectiveness of GNN-based EBMs for tree topology estimation and (ii) verifying the improvement of GNN-based branch length parameterization for VBPI over the baseline approaches (i.e., split and PSP based parameterizations) and investigating how helpful the learnable topological features are for reducing the amortization gaps.

## 4.1   SIMULATED DATA TREE PROBABILITY ESTIMATION

We first investigated the representative power of learnable topological features for approximating distributions on phylogenetic trees using energy based models (EBMs), and conducted experiments on a simulated data set. We used the space of unrooted phylogenetic trees with 8 leaves, which contains 10395 unique trees in total. Similarly as in Zhang & Matsen IV (2019), we generated a target distribution $p_0(\tau)$ by drawing a sample from the symmetric Dirichlet distribution $\mathrm{Dir}(\beta\mathbf{1})$ of order 10395 with a pre-selected arbitrary order of trees. The concentration parameter $\beta$ is used to control the diffuseness of the target distribution and was set to 0.008 to provide enough information for inference while allowing for adequate diffusion in the target. As mentioned earlier in section 3.2.1, we used noise contrastive estimation (NCE) to train our EBMs where we set the noise distribution $p_n(\tau)$ to be the uniform distribution. Results were collected after 200,000 parameter updates. Note that the minimum NCE loss in this case is

$$J^* = -2\mathrm{JSD}\left(p_0(\tau)\|p_n(\tau)\right) + 2\log 2,$$

where $\mathrm{JSD}(\cdot\|\cdot)$ is the Jensen-Shannon divergence.

Figure 2 shows the empirical performance of different methods. From the left plot, we see that the NCE losses converge rapidly and the gaps between NCE losses for the GNN variants and the best NCE loss $J^*$ (dashed red line) are close to zero, demonstrating the representative power of learnable topological features on phylogenetic tree probability estimations. The evolution of KL divergences (middle plot) is consistent with the NCE losses. Compared to MLP, all GNN variants perform better, indicating that the extra flexibility provided by GNN iterations is crucial for tree probability estimation that would benefit from more informative graph-level features. Although the

Table 1: Evidence Lower bound (ELBO) and marginal likelihood (ML) estimates of different methods across 8 benchmark datasets for Bayesian phylogenetic inference. The marginal likelihood estimates of all variational methods are obtained via importance sampling using 1000 samples, and the results (in units of nats) are averaged over 100 independent runs with standard deviation in brackets. Results for stepping-stone (SS) are from Zhang & Matsen IV (2019)(using 10 independent MrBayes (Ronquist et al., 2012) runs, each with 4 chains for 10,000,000 iterations and sampled every 100 iterations).

| | DATA SET | DS1 | DS2 | DS3 | DS4 | DS5 | DS6 | DS7 | DS8 |
|---|---|---|---|---|---|---|---|---|---|
| | # TAXA | 27 | 29 | 36 | 41 | 50 | 50 | 59 | 64 |
| | # SITES | 1949 | 2520 | 1812 | 1137 | 378 | 1133 | 1824 | 1008 |
| ELBO | SPLIT | -7112.16(2.00) | -26369.95(0.89) | -33736.87(0.36) | -13332.82(0.75) | -8218.86(0.22) | -6729.49(0.47) | -37335.44(0.12) | -8661.56(2.00) |
| | PSP | -7111.23(1.21) | -26369.43(0.64) | -33736.71(0.45) | -13332.42(0.65) | -8218.34(0.16) | -6729.20(0.44) | -37335.18(0.13) | -8655.40(0.32) |
| | MLP | -7110.43(0.10) | -26368.85(0.10) | -33736.25(0.05) | -13331.99(0.09) | -8218.23(0.12) | -6728.98(0.17) | -37334.99(0.12) | -8655.33(0.15) |
| | GCN | -7110.32(0.10) | -26368.88(0.09) | -33736.25(0.05) | -13331.94(0.09) | -8218.06(0.10) | -6728.78(0.16) | -37334.92(0.12) | -8655.17(0.15) |
| | GIN | -7110.27(0.09) | -26368.86(0.09) | -33736.26(0.07) | -13331.82(0.09) | -8217.88(0.11) | -6728.59(0.17) | -37334.94(0.11) | **-8655.00(0.15)** |
| | SAGE | -7110.29(0.10) | **-26368.84(0.07)** | -33736.28(0.06) | -13331.84(0.10) | -8217.92(0.11) | -6728.63(0.15) | -37334.91(0.11) | -8655.05(0.14) |
| | GGNN | **-7110.26(0.10)** | **-26368.84(0.10)** | **-33736.20(0.06)** | **-13331.79(0.09)** | -8217.88(0.11) | **-6728.56(0.16)** | -37334.87(0.12) | -8655.01(0.15) |
| | EDGE | **-7110.26(0.10)** | **-26368.84(0.09)** | -33736.25(0.08) | -13331.80(0.10) | **-8217.80(0.12)** | -6728.57(0.16) | **-37334.84(0.14)** | -8655.01(0.14) |
| ML | SPLIT | -7108.47(0.27) | -26367.73(0.08) | -33735.12(0.12) | -13330.01(0.31) | -8214.83(0.51) | -6724.58(0.48) | -37332.18(0.43) | -8651.39(0.94) |
| | PSP | -7108.41(0.19) | -26367.74(0.09) | -33735.12(0.10) | -13329.96(0.22) | -8214.66(0.46) | -6724.41(0.49) | -37332.05(0.33) | -8650.66(0.51) |
| | MLP | -7108.41(0.24) | -26367.74(0.08) | -33735.12(0.10) | -13329.99(0.22) | -8214.77(0.47) | -6724.47(0.47) | -37332.03(0.34) | -8650.72(0.53) |
| | GCN | -7108.43(0.16) | -26367.73(0.08) | -33735.12(0.11) | -13329.97(0.22) | -8214.69(0.45) | -6724.51(0.47) | -37332.04(0.29) | -8650.68(0.54) |
| | GIN | -7108.40(0.22) | -26367.73(0.08) | -33735.12(0.10) | -13329.96(0.18) | -8214.64(0.39) | **-6724.41(0.40)** | -37332.02(0.30) | **-8650.66(0.45)** |
| | SAGE | -7108.40(0.17) | **-26367.73(0.07)** | **-33735.12(0.09)** | **-13329.96(0.17)** | -8214.64(0.44) | -6724.39(0.42) | -37332.01(0.30) | -8650.66(0.49) |
| | GGNN | -7108.40(0.19) | -26367.73(0.10) | **-33735.11(0.09)** | -13329.95(0.19) | -8214.67(0.36) | -6724.38(0.42) | -37332.03(0.30) | -8650.68(0.48) |
| | EDGE | **-7108.41(0.14)** | **-26367.73(0.07)** | **-33735.12(0.09)** | -13329.94(0.19) | -8214.64(0.38) | **-6724.37(0.40)** | **-37332.04(0.26)** | **-8650.65(0.45)** |
| | SS | -7108.42(0.18) | -26367.57(0.48) | -33735.44(0.50) | -13330.06(0.54) | **-8214.51(0.28)** | -6724.07(0.86) | -37332.76(2.42) | -8649.88(1.75) |

raw features from interior node embedding contain all information of phylogenetic tree topologies, we see that distilling effective structural information from them is still challenging. This makes GNN models that are by design more capable of learning geometric representations a favorable choice. The right plot compares the probability mass approximations provided by EBMs using MLP and GGNN (which performs the best among all GNN variants), to the ground truth $p_0(\tau)$. We see that EBMs using GGNN consistently provide accurate approximations for trees across a wide range of probabilities. On the other hand, estimates provided by those using MLP are often of large bias, except for a few trees with high probabilities.

## 4.2 REAL DATA VARIATIONAL BAYESIAN PHYLOGENETIC INFERENCE

The second task we considered is VBPI, where we compared learnable topological feature based branch length parameterizations to heuristic feature based parameterizations (denoted as Split and PSP resepectively) proposed in the original VBPI approach (Zhang & Matsen IV, 2019). All methods were evaluated on 8 real datasets that are commonly used to benchmark Bayesian phylogenetic inference methods (Hedges et al., 1990; Garey et al., 1996; Yang & Yoder, 2003; Henk et al., 2003; Lakner et al., 2008; Zhang & Blackwell, 2001; Yoder & Yang, 2004; Rossman et al., 2001; Höhna & Drummond, 2012; Larget, 2013; Whidden & Matsen IV, 2015). These datasets, which we call DS1-8, consist of sequences from 27 to 64 eukaryote species with 378 to 2520 site observations. We concentrate on the most challenging part of the Bayesian phylogenetics: joint learning of the tree topologies and the branch lengths, and assume a uniform prior on the tree topology, an i.i.d. exponential prior ($\text{Exp}(10)$) for the branch lengths and the simple Jukes & Cantor (1969) substitution model. We gathered the support of CPTs from 10 replicates of 10000 ultrafast maximum likelihood bootstrap trees (Minh et al., 2013). We set $K = 10$ for the multi-sample lower bound, with a schedule $\lambda_n = \min(1, 0.001 + n/100000)$, going from 0.001 to 1 after 100000 iterations. The Monte Carlo gradient estimates for the tree topology parameters and branch length parameters were obtained via VIMCO (Mnih & Rezende, 2016) and the reparameterization trick (Kingma & Welling, 2014) respectively. Results were collected after 400,000 parameter updates.

Table 1 shows the estimates of the evidence lower bound (ELBO) and the marginal likelihood using different branch length parameterizations on the 8 benchmark datasets, including the results for the

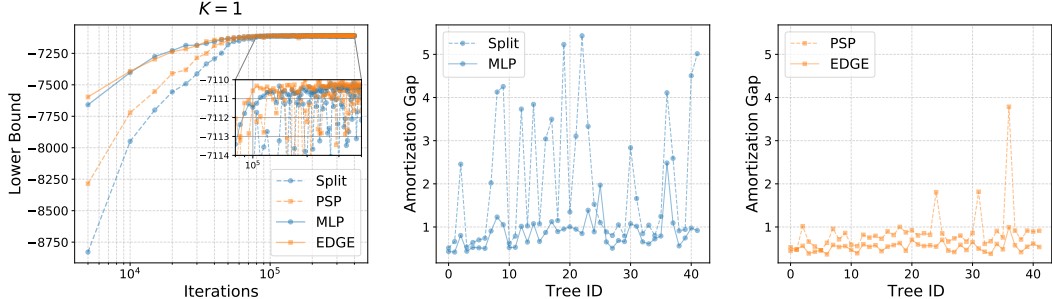

Figure 3: Performance on DS1. **Left:** Lower bounds. **Middle & Right:** Amortization gaps on trees in the 95% credible sets.

stepping-stone (SS) method (Xie et al., 2011), which is one of the state-of-the-art sampling based methods for marginal likelihood estimation. For each data set, a better approximation would lead to a smaller variance of the marginal likelihood estimates. We see that solely using the raw features, MLP-based parameterization already outperformed the Split and PSP baselines by providing tighter lower bounds. With more expressive representations of local structures enabled by GNN iterations, GNN-based parameterization further improved upon MLP-based methods, indicating the importance of harnessing local topological information for flexible branch length distributions. Moreover, when used as importance distributions for marginal likelihood estimation via importance sampling, MLP and GNN variants provide more steady estimates (less variance) than Split and PSP respectively. All variational approaches compare favorably to SS and require much fewer samples. The left plot in Figure 3 shows the evidence lower bounds as a function of the number of parameter updates on DS1. Although neural networks based parameterization adds to the complexity of training in VI, we see that by the time Split and PSP converge, MLP and EDGE[3] achieve comparable (if not better) lower bounds and quickly surpass these baselines as the number of iteration increases.

As diagonal Lognormal branch length distributions were used for all parameterization methods, how these variational distributions were amortized over tree topologies under different parameterizations therefore is crucial for the overall approximation performance. To better understand this effect of amortized inference, we further investigated the amortization gaps[4] of different methods on individual trees in the 95% credible set of DS1 as in Zhang (2020). The middle and right plots in Figure 3 show the amortization gaps of different parameterization methods on each tree topology $\tau$. We see the amortization gaps of MLP and EDGE are considerably smaller than those of Split and PSP respectively, showing the efficiency of learnable topological features for amortized branch length distributions. Again, incorporating more local topological information is beneficial, as evidenced by the significant improvement of EDGE over MLP. More results about the amortization gaps can be found in Table 2 in the appendix.

## 5 CONCLUSION

We presented a novel approach for phylogenetic inference based on learnable topological features. By combining the raw node features that minimize the Dirichlet energy with modern GNN variants, our learnable topological features can provide efficient structural information without requiring domain expertise. In experiments, we demonstrated the effectiveness of our approach for tree probability estimation on simulated data and showed that our method consistently outperforms the baseline approaches for VBPI on a benchmark of real data sets. Future work would investigate more sophisticated GNNs for phylogenetic trees, and applications to other phylogenetic inference tasks where efficiently leveraging structural information of tree topologies is of great importance.

---

[3]We use EDGE as an example here for branch length parameterization since it can learn edge features (see Appendix F). All GNN variants (except the simple GCN) performed similarly in this example (see Table 1).

[4]The amortization gap on a tree topology $\tau$ is defined as $L(Q^*|\tau) - L(Q_\psi|\tau)$, where $L(Q_\psi|\tau)$ is the ELBO of the approximating distribution $Q_\psi(q|\tau)$ and $L(Q^*|\tau)$ is the maximum lower bound that can be achieved with the same variational family. See more details in Zhang (2020); Cremer et al. (2018).

ACKNOWLEDGMENTS

This work was supported by National Natural Science Foundation of China (grant no. 12201014), as well as National Institutes of Health grant AI162611. The research of the author was support in part by the Key Laboratory of Mathematics and Its Applications (LMAM) and the Key Laboratory of Mathematical Economics and Quantitative Finance (LMEQF) of Peking University. The author is grateful for the computational resources provided by the High-performance Computing Platform of Peking University. The author appreciates the anonymous ICLR reviewers for their constructive feedback.

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

# A  SUBSPLIT BAYESIAN NETWORKS

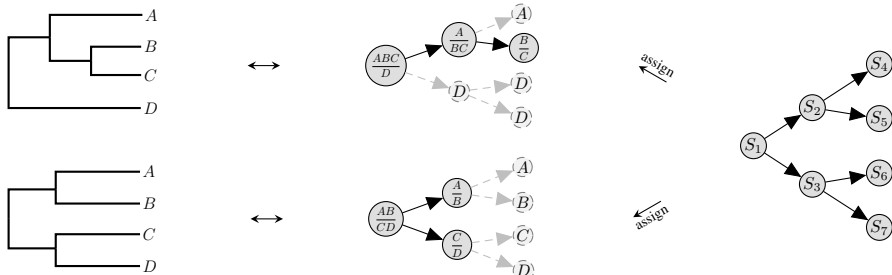

Figure 4: A simple subsplit Bayesian network for a leaf set that contains 4 species A, B, C and D. **Left**: Some rooted phylogenetic tree examples. **Middle**: The corresponding SBN assignments. For ease of illustration, subsplit $(W, Z)$ is represented as $\frac{W}{Z}$ in the graph. The *dashed gray subgraphs* represent fake splitting processes where splits are deterministically assigned, and are used purely to complement the networks such that the overall network has a fixed structure. **Right**: The SBN for these examples. This figure is adapted from Zhang & Matsen IV (2019).

Subsplit Bayesian networks (SBNs) were introduced by Zhang & Matsen IV (2018) for density estimation on tree topologies and were later used to provide a family of variational distributions for Bayesian phylogenetic inference (Zhang & Matsen IV, 2019). Given a leaf set $\mathcal{X}$ of size $N$, one can define a subsplit Bayesian network $B_{\mathcal{X}}$ to be a Bayesian network whose nodes take on subsplit or singleton clade values that represent the local topological structures of trees (Figure 4). This formulation allows us to represent rooted tree topologies as SBN assignments. More specifically, one can follow the splitting process (see the *solid dark subgraphs* in Figure 4, middle) of the tree and assign the subsplits to the corresponding nodes along the way to get a unique subsplit decomposition of the tree topology. Given the subsplit decomposition of a rooted tree $\tau = \{s_1, s_2, \ldots\}$, where $s_1$ is the root subsplit, the SBN-induced tree probability of $\tau$ is

$$p_{\text{sbn}}(T = \tau) = p(S_1 = s_1) \prod_{i>1} p(S_i = s_i | S_{\pi_i} = s_{\pi_i})$$

where $S_i$ denote the subsplit- or singleton-clade-valued random variables at node $i$ and $\pi_i$ is the index set of the parents of $S_i$. As Bayesian networks, SBN-induced distributions are all naturally normalized. We can also adjust the structures of SBNs for a wide range of expressive distributions, as long as they remain valid directed acyclic graphs (DAGs). In practice, the simplest SBN (the one with a full and complete binary tree structure as shown in Figure 4) is often found to be good enough.

The SBN framework also generalizes to unrooted trees, which are the most common type of phylogenetic trees. The key is to view unrooted trees as rooted trees with missing roots. Marginalizing out the unobserved root nodes leads to the SBN probability estimates for unrooted trees

$$p_{\text{sbn}}(T^{\text{u}} = \tau) = \sum_{s_1 \sim \tau} p(S_1 = s_1) \prod_{i>1} p(S_i = s_i | S_{\pi_i} = s_{\pi_i})$$

where $\sim$ means all root subsplits that are compatible with $\tau$ (i.e., root subsplits of the edges of $\tau$).

We can evaluate the SBN probabilities of tree topologies efficiently through a two pass algorithm (Zhang & Matsen IV, 2018). As Bayesian networks, SBNs also allow fast sampling procedures (i.e., ancestral sampling). These properties make SBNs a natural choice for variational inference.

Parameterizing SBNs in VBPI requires a sufficiently large subsplit *support* of CPTs (i.e., where the associate conditional probabilities are allowed to take nonzero values) that covers favorable parent child subsplit pairs from trees with high posterior probabilities. In practice, a simple bootstrap-based approach has been found effective for providing such a support (Zhang & Matsen IV, 2019). Let $\mathbb{S}_{\text{r}}$ denote the set of root subsplits (e.g., the splits) in the support and $\mathbb{S}_{\text{ch}|\text{pa}}$ denote the set of parent-child subsplit pairs in the support. The CPTs can be defined via the softmax function as follows

$$p(S_1 = s_1) = \frac{\exp(\phi_{s_1})}{\sum_{s_{\text{r}} \in \mathbb{S}_{\text{r}}} \exp(\phi_{s_{\text{r}}})}, \quad p(S_i = s | S_{\pi_i} = t) = \frac{\exp(\phi_{s|t})}{\sum_{s \in \mathbb{S}_{\cdot|t}} \exp(\phi_{s|t})}.$$

## B THE TWO-PASS ALGORITHM FOR INTERIOR NODE EMBEDDING

Due to the hierarchical structure of phylogenetic trees, we derive a linear time two-pass algorithm for solving the linear equations 4. In the first pass, we traverse the tree in a postorder fashion and express the node features as a linear function of those of their parents as follows

$$\widehat{\boldsymbol{x}}_u = c_u \widehat{\boldsymbol{x}}_{\pi_u} + \boldsymbol{d}_u,$$

for all the nodes except the root node, where $\pi_u$ is the parent node of $u$. For all leaf nodes $u \in V^b$, this is straightforward: $c_u = 0$, $\boldsymbol{d}_u = \boldsymbol{x}_u$. For any interior node $u$ that is not the root node, as a result of postorder traversal, $c_v, \boldsymbol{d}_v$ is available $\forall v \in \mathrm{ch}(u)$ when $u$ is visited. Therefore, we can rewrite the equation about $u$ in 4 as

$$|\mathcal{N}(u)| \widehat{\boldsymbol{x}}_u = \widehat{\boldsymbol{x}}_{\pi_u} + \sum_{v \in \mathrm{ch}(u)} \widehat{\boldsymbol{x}}_v$$

$$= \widehat{\boldsymbol{x}}_{\pi_u} + \sum_{v \in \mathrm{ch}(u)} (c_v \widehat{\boldsymbol{x}}_u + \boldsymbol{d}_v).$$

This implies

$$\widehat{\boldsymbol{x}}_u = \frac{1}{|\mathcal{N}(u)| - \sum_{v \in \mathrm{ch}(u)} c_v} \cdot \widehat{\boldsymbol{x}}_{\pi_u} + \frac{\sum_{v \in \mathrm{ch}(u)} \boldsymbol{d}_v}{|\mathcal{N}(u)| - \sum_{v \in \mathrm{ch}(u)} c_v},$$

which gives the recursive updating formula in equation 6.

In the second pass, we traverse the tree in a preorder fashion. We first visit the root node $r$. From the first pass, $c_v, \boldsymbol{d}_v$ is available $\forall v \in \mathrm{ch}(r)$. Similarly, from equation 4, we have

$$|\mathcal{N}(r)| \widehat{\boldsymbol{x}}_r = \sum_{v \in \mathrm{ch}(r)} \widehat{\boldsymbol{x}}_v$$

$$= \sum_{v \in \mathrm{ch}(r)} (c_v \widehat{\boldsymbol{x}}_r + \boldsymbol{d}_v)$$

Therefore,

$$\widehat{\boldsymbol{x}}_r = \frac{\sum_{v \in \mathrm{ch}(r)} \boldsymbol{d}_v}{|\mathcal{N}(r)| - \sum_{v \in \mathrm{ch}(r)} c_v}.$$

For any other interior node $u$, as a result of preorder traversal, $\widehat{\boldsymbol{x}}_{\pi_u}$ is available when $u$ is visited. We can compute its node feature $\widehat{\boldsymbol{x}}_u$ via equation 5.

## C PROOFS FOR LEMMAS, COROLLARIES, AND THEOREMS

**Proof for Lemma 1**

**Lemma 1.** *Let $\lambda = \min_{u \in V^o \setminus \{r\}} |\mathcal{N}(u)|$. For all interior node $u \in V^o \setminus \{r\}$, $0 \le c_u \le \frac{1}{\lambda-1}$.*

*Proof.* We prove by induction. Since $|\mathcal{N}(u)| \ge 2$, $\forall u \in V^o$, we have $\lambda \ge 2$. Note that $c_u = 0$, $\forall u \in V^b$. Suppose $0 \le c_v \le \frac{1}{\lambda-1}$, $\forall v \in \mathrm{ch}(u)$, it suffices to show that $0 \le c_u \le \frac{1}{\lambda-1}$ as long as $u$ is not the root node. From the recursive updating formula, we have

$$0 \le c_u = \frac{1}{|\mathcal{N}(u)| - \sum_{v \in \mathrm{ch}(u)} c_v} \le \frac{1}{|\mathcal{N}(u)| - \sum_{v \in \mathrm{ch}(u)} \frac{1}{\lambda-1}} = \frac{1}{|\mathcal{N}(u)| - \frac{|\mathcal{N}(u)|-1}{\lambda-1}}.$$

As $2 \le \lambda \le |\mathcal{N}(u)|$,

$$(|\mathcal{N}(u)| - \lambda) \frac{\lambda-2}{\lambda-1} \ge 0 \Rightarrow |\mathcal{N}(u)| - \lambda \ge \frac{|\mathcal{N}(u)| - \lambda}{\lambda-1} \Rightarrow |\mathcal{N}(u)| - \frac{|\mathcal{N}(u)|-1}{\lambda-1} \ge \lambda - 1.$$

Therefore, $0 \le c_u \le \frac{1}{\lambda-1}$. $\square$

**Remark.** For bifurcating phylogenetic trees, we have $\lambda = 3 \Rightarrow 0 \le c_u \le \frac{1}{2}$.

**Proof for Lemma 2**

**Lemma 2** (Extremum Principle). *Let $\{\widehat{\boldsymbol{x}}_u \in \mathbb{R}^d | u \in V\}$ be a set of d-dimensional node features that satisfies equations 4. $\forall 1 \leq n \leq d$, let $\widehat{\boldsymbol{X}}[n] = \{\widehat{\boldsymbol{x}}_u[n] | u \in V\}$ be the set of the n-th components of node features. Then, $\forall 1 \leq n \leq d$, we have: (i) the extremum values (i.e., maximum and minimum) of $\widehat{\boldsymbol{X}}[n]$ can be achieved at some tip nodes; (ii) if the extremum values are achieved at some interior nodes, then $\widehat{\boldsymbol{X}}[n]$ has only one member, or in other words, $\widehat{\boldsymbol{x}}_u[n]$ is the same $\forall u \in V$.*

*Proof.* From equations 4, we have

$$\widehat{\boldsymbol{x}}_u = \frac{1}{|\mathcal{N}(u)|} \sum_{v \in \mathcal{N}(u)} \widehat{\boldsymbol{x}}_v, \quad \forall u \in V^o.$$

In other words, for any interior node, its node feature is the mean of those of its neighbors. Therefore, $\forall 1 \leq n \leq d$, the extremum values of $\widehat{\boldsymbol{X}}[n]$ can be achieved at the boundary of the graph, i.e., the tip nodes. On the other hand, if the extremum value of $\widehat{\boldsymbol{X}}[n]$ is achieved at some interior node $u$, then the extremum is also achieved at all the neighbors of $u$. Since the tree topology is connected, this implies that $\widehat{\boldsymbol{x}}_u[n]$ is a constant $\forall u \in V$. □

**Proof for Theorem 1**

**Theorem 1.** *Let $N$ be the number of tip nodes. Let $\{\widehat{\boldsymbol{x}}_u \in \mathbb{R}^N | u \in V\}$ be the solution to the linear system 4 with one hot encoding for the tip nodes. Then, $\forall u \in V^o$, we have*

$$(i)\ 0 < \widehat{\boldsymbol{x}}_u[n] < 1, \quad \forall 1 \leq n \leq N, \quad and \quad (ii)\ \sum_{n=1}^{N} \widehat{\boldsymbol{x}}_u[n] = 1.$$

*Proof.* As the tip node features are one hot vectors, $(i)$ follows immediately from Lemma 2. Let $S = \{\boldsymbol{x} \in \mathbb{R}^N | \sum_{n=1}^{N} \boldsymbol{x}[n] = 1\}$ and $P_S$ be the projection onto $S$. Since $\{\widehat{\boldsymbol{x}}_u \in \mathbb{R}^N | u \in V\}$ solves the linear system 4, it minimizes the Dirichlet energy. Now consider its projection $\{P_S(\widehat{\boldsymbol{x}}_u) \in \mathbb{R}^N | u \in V\}$. Since one hot vectors are in $S$, $P_S(\widehat{\boldsymbol{x}}_u) = \widehat{\boldsymbol{x}}_u, \forall u \in V^b$. Note that $P_S$ is a projection operator, we have

$$\sum_{(u,v) \in E} \|P_S(\widehat{\boldsymbol{x}}_u) - P_S(\widehat{\boldsymbol{x}}_v)\|^2 \leq \sum_{(u,v) \in E} \|\widehat{\boldsymbol{x}}_u - \widehat{\boldsymbol{x}}_v\|^2.$$

Note that $\{\widehat{\boldsymbol{x}}_u \in \mathbb{R}^N | u \in V\}$ minimizes the Dirichlet energy, the equality has to hold which implies that $\widehat{\boldsymbol{x}}_u - \widehat{\boldsymbol{x}}_v$ is parallel to the hyperplane $S$, $\forall (u,v) \in E$. Therefore, $\widehat{\boldsymbol{x}}_u \in S, \forall u \in V^o$ as the tree topology is connected and the tip node features are already in $S$. □

**Proof for Corollary 1**

**Corollary 1.** *Suppose that the tip node features are linearly independent, the interior node features obtained from the solution to the linear system 4 all lie in the interior of the convex hull of all tip node features.*

*Proof.* Let $\boldsymbol{A}$ be a matrix whose columns corresponds to the tip node features. From the linear system 4, we see that all interior node features lie in the column space of $\boldsymbol{A}$ and hence can be uniquely represented as

$$\widehat{\boldsymbol{x}}_u = \boldsymbol{A}\widehat{\boldsymbol{y}}_u, \quad \forall u \in V$$

as the columns are linearly independent. Moreover, $\{\widehat{\boldsymbol{y}}_u \in \mathbb{R}^N | u \in V\}$ satisfies the same linear system 4

$$\sum_{v \in \mathcal{N}(u)} (\widehat{\boldsymbol{y}}_u - \widehat{\boldsymbol{y}}_v) = \boldsymbol{0}, \quad \forall u \in V^o, \qquad \widehat{\boldsymbol{y}}_v = \boldsymbol{y}_v, \quad \forall v \in V^b,$$

where $\{\boldsymbol{y}_v \in \mathbb{R}^N | v \in V^b\}$ are one hot encoding vectors. By Theorem 1, we have $\forall u \in V^o$

$$(i)\ 0 < \widehat{\boldsymbol{y}}_u[n] < 1, \quad \forall 1 \leq n \leq N, \quad and \quad (ii)\ \sum_{n=1}^{N} \widehat{\boldsymbol{y}}_u[n] = 1.$$

which concludes the proof. □

**Proof for Lemma 3**

**Lemma 3.** *Let $\{\widehat{\boldsymbol{x}}_u | u \in V\}$ be the solution to the linear system 4, with linearly independent tip node features. Let $\{\widehat{\boldsymbol{x}}_u = \sum_{v \in V^b} a_u^v \boldsymbol{x}_v | u \in V^o\}$ be the convex combination representations of the interior node features. For any tip node $v \in V^b$, we have*

$$u^* = \arg\max_{u \in V^o} a_u^v \quad \Leftrightarrow \quad u^* \in \mathcal{N}(v).$$

*Proof.* For any tip node $v \in V^b$, let $\bar{v} \in \mathcal{N}(v)$ be the adjacent interior node of $v$. It suffices to show that $a_{\bar{v}}^v > a_u^v, \forall u \in V^o \backslash \{\bar{v}\}$. As $\{\boldsymbol{x}_v | v \in V^b\}$ are linearly independent, from equations 4 we have

$$a_u^v = \frac{\sum_{w \in \mathcal{N}(u)} a_w^v}{|\mathcal{N}(u)|}, \quad \forall u \in V^o. \tag{11}$$

Now consider a new tree topology $\tau'$ that has $v$ removed from $\tau$. Note that equation 11 still holds for all interior nodes except $\bar{v}$, the maximum value of $\{a_u^v | u \in V \backslash \{v\}\}$ therefore can be achieved at either the boundary of $\tau'$ or $\bar{v}$. As $a_u^v = 0, \forall u \in V^b \backslash \{v\}$ and $a_{\bar{v}}^v > 0$ (by Corollary 1), the maximum value hence is achieved at $a_{\bar{v}}^v$, i.e., $a_{\bar{v}}^v \geq a_u^v, \forall u \in V^o$. Now suppose there exists an interior node $u \neq \bar{v}$ such that $a_u^v = a_{\bar{v}}^v$. Similarly to Lemma 2, this implies $a_u^v = a_{\bar{v}}^v, \forall u \in V \backslash \{v\}$, which is an obvious contradiction. $\square$

**Proof for Theorem 2**

**Theorem 2** (Identifiability). *Let $\boldsymbol{X}^o = \{\widehat{\boldsymbol{x}}_u | u \in V^o\}$ and $\boldsymbol{Z}^o = \{\widehat{\boldsymbol{z}}_u | u \in V^o\}$ be the sets of interior node features that minimizes the Dirichlet energy for phylogenetic tree topologies $\tau_x$ and $\tau_z$ respectively, given the same linearly independent tip node features. If $\boldsymbol{X}^o = \boldsymbol{Z}^o$, then $\tau_x = \tau_z$.*

*Proof.* Consider the case for unrooted trees first. We prove by induction on the number of tip nodes $N$ of the tree topology. For $N = 3$, the tree topology is trivial. If $N > 3$, then for each tip node, Lemma 3 identifies its adjacent node by convex combination coefficient maximization. As the number of interior nodes is less than the number of tip nodes, there must be two tip nodes that connect to the same interior node. Merging the two tip nodes into their shared neighbor reduces the problem to size $N - 1$, with the shared neighbor being the new tip node. It is easy to check that for this new set of node features, the tip node features are also linearly independent. Therefore, the remaining part of the tree topology can be recovered by induction hypothesis. The proof for rooted trees is similar. $\square$

## D   MORE RESULTS ON AMORTIZATION GAPS

Table 2: Amortization gaps on trees in the 95% credible set of DS1.

|          | SPLIT | PSP  | MLP  | GCN  | GIN  | SAGE | GGNN | EDGE |
|----------|-------|------|------|------|------|------|------|------|
| TREE 24  | 1.53  | 1.81 | 0.89 | 1.28 | 0.56 | **0.52** | 0.54 | 0.55 |
| TREE 31  | 1.66  | 1.82 | 1.02 | 1.29 | 0.61 | 0.63 | 0.56 | **0.50** |
| TREE 36  | 4.11  | 3.79 | 2.48 | 2.31 | 1.68 | 1.28 | 1.25 | **1.00** |
| AVERAGE  | 2.06  | 0.89 | 0.87 | 0.94 | 0.63 | 0.63 | 0.57 | **0.53** |

## E   RELATED WORKS

Many deep learning approaches have also been proposed for phylogenetic models recently. Solis-Lemus et al. (2022) use symmetry-preserving neural networks to explicitly encode permutation invariance of phylogenetic trees. However, their method requires enumeration of all permutations of tip nodes that would leave tree topologies unchanged and hence is challenging to extend to datasets with many sequences. Jiang et al. (2022) learn hyperbolic embeddings of gene sequences for phylogenetic tree placement and updates. While these embeddings maybe useful for sequentially updating species trees with only a handful of genes, they are not suitable for learning representations

of tree topologies and their local structures with a given taxa set. Fioravanti et al. (2018) use CNNs to incorporate phylogenetic information for the classification of metagenomics data where the phylogenetic tree is assumed to be known. Their method, therefore, is also not suitable for phylogenetic inference where the tree topology is unknown and needs to be inferred from the data.

## F    DETAILS ON GRAPH CONVOLUTIONAL OPERATORS

The followings are the update and aggregate functions for the graph convolutional operators used in our experiments.

- Graph convolutional networks (GCN)

$$h_v^{(t+1)} = W^{(t)} \frac{m_v^{(t+1)}}{\sqrt{1 + d_v}}, \quad m_v^{(t+1)} = \sum_{u \in \mathcal{N}(v) \cup \{v\}} \frac{h_u^{(t)}}{\sqrt{1 + d_u}},$$

where $d_u$ is the degree of node $u$.

- Graph isomorphism networks (GIN)

$$h_v^{(t+1)} = \text{MLP}^{(t)} \left( (1 + \epsilon^{(t)}) h_v^{(t)} + m_v^{(t+1)} \right), \quad m_v^{(t+1)} = \sum_{u \in \mathcal{N}(v)} h_u^{(t)},$$

where $\epsilon^{(t)}$ can be either a learnable parameter or a fixed scalar.

- GraphSAGE operator (SAGE)

$$h_v^{(t+1)} = W_1^{(t)} h_v^{(t)} + W_2^{(t)} m_v^{(t+1)}, \quad m_v^{(t+1)} = \frac{\sum_{u \in \mathcal{N}(v)} h_u^{(t)}}{d_v}.$$

- Gated graph convolutional operator (GGNN)

$$h_v^{(t+1)} = \text{GRU}^{(t)} \left( m_v^{(t+1)}, h_v^{(t)} \right), \quad m_v^{(t+1)} = \sum_{u \in \mathcal{N}(v)} W^{(t)} h_u^{(t)},$$

where $\text{GRU}^{(t)}$ is a Gated recurrent unit, a gating mechanism in recurrent neural networks.

- Edge convolutional operator (EDGE)

$$h_v^{(t+1)} = \sum_{u \in \mathcal{N}(v)} e_{u \to v}^{(t+1)}, \quad e_{u \to v}^{(t+1)} = \text{MLP}^{(t)} \left( h_v^{(t)} \parallel h_u^{(t)} - h_v^{(t)} \right), \ \forall u \in \mathcal{N}(v)$$

where $\parallel$ means concatenation and $e_{u \to v}^{(t+1)}$ is the edge feature from node $u$ to node $v$.

