# OpenReview forum: "Learnable Topological Features For Phylogenetic Inference via Graph Neural Networks"
_ICLR.cc/2023/Conference — ICLR 2023 poster_

### Official Review · Reviewer_uXSY · 2022-10-24

**Confidence:** 3
**Correctness:** 3
**Technical Novelty And Significance:** 2
**Empirical Novelty And Significance:** 2
**Recommendation:** 6

**Clarity, Quality, Novelty And Reproducibility:**

The presentation of the paper can be improved. While I believe the idea of this paper is relatively novel, the paper should provide more clarity on the background the models for the ICLR audiences.

**Strength And Weaknesses:**

Strength:
The paper studies an interesting problem of phylogenetic inference. The author adopted graph neural networks to help learn the structure information of phylogenetic trees, and perform inference on downstream tasks.

Weakness:
I found the paper hard to read. While I understand that the authors might expects some knowledge of phylogenetics, it is necessary to provide a good background and clear explanations of the notations. For example, does the paper focus on rooted phylogenetic trees or including the unrooted one? The illustration figure showed an unrooted phylogenetic tree, while some discussions assume a root node?

The concepts and notations are also confusing. For example, in the phylogenetic model description, the author use Y to represent some "observed sequences", but the concept was never mentioned before. Also, what are the "characters" here refer to? The author never explained what "sites" refers to, but gives the assumption "different sites are independent and identically distributed". It would be good to explain what it means.

In the proposed method section, it is hard to tell which part describes the existing work and which part is proposed by the authors. And if I understand correctly, the learning of node features based on the Dirichlet energy does require the knowledge of the phylogenetic tree structure. However, I thought the original problem is to find the appropriate topological structure?

I also found the experiment results not convincing enough. First, the author tested multiple GNN algorithms to compare with the results. The authors need to consider the multiple hypothesis testing effect. Second, in table 1, many results are not significant even though it is highlighted in bold. The difference between the proposed method with the original SPLIT and PSP is quite small in most cases.


Minor comment:
There is a type after equation (1). I believe the root node is \tao instead of r?

**Summary Of The Paper:**

The paper studies the problem of phylogenetic inference. The authors proposed a structural representation method for phylogenetic inference based on learnable topological features. The learnable topological features are constructed by minimizing the Dirichlet energy. The authors use the combined node features to infer structural information of phylogenetic trees and for downstream tasks. The authors conducted experiments on a simulated data and a benchmark of real data variational Bayesian phylogenetic inference problem to demonstrate the effectiveness of the proposed method.

**Summary Of The Review:**

The authors presented a novel idea to use graph learning for phylogenetic analysis. The authors claimed that their learned features has good representation power of the tree topology, and conducted some theoretical analysis on the representation power. The authors did experiments with several GNN algorithms, but the results look not strong enough (and interestingly GIN does not have much advantage in this case). More clarification on the background and method is also needed for this paper.

---

> ### Author Response · Authors · 2022-11-10
> **Response to Reviewer uXSY (Part 1/2)**
>
> Thank you for your review and feedback. We address your specific questions and comments as follows
>
> Q1: I found the paper hard to read. While I understand that the authors might expects some knowledge of phylogenetics, it is necessary to provide a good background and clear explanations of the notations. For example, does the paper focus on rooted phylogenetic trees or including the unrooted one? The illustration figure showed an unrooted phylogenetic tree, while some discussions assume a root node?
>
> A1: Thanks for raising this up. The method developed in this paper works for both rooted and unrooted trees. However, in the experiment section, we focus on unrooted trees. There are two places where we indeed used root nodes in our discussion. One is for the phylogenetic likelihood computation. For unrooted trees, the phylogenetic likelihood is computed using a virtual root node (which can be any interior node given a reversible Markov substitution model). The other is for the tree traversal when introducing the two-pass algorithm for Dirichlet energy minimization. For unrooted trees, this is usually done by choosing an interior node as a virtual root node and doing the traversals afterwards (we have a footnote explanation of this in our paper). We will clarify this and add more background and explanations in our revision.
>
> Q2: The concepts and notations are also confusing. For example, in the phylogenetic model description, the author use Y to represent some "observed sequences", but the concept was never mentioned before. Also, what are the "characters" here refer to? The author never explained what "sites" refers to, but gives the assumption "different sites are independent and identically distributed". It would be good to explain what it means.
>
> A2: The observed sequences are the molecular data used in phylogenetic inference. They are usually aligned DNA,RNA, or protein sequences for the observed species of interest. We have indeed mentioned it several times before introducing the notation of $Y$. The characters refer to the values each entry in the sequences can take on. For example, if the data are DNA sequences, then the characters would be the nucleotides A, G, C, T (DNA bases). The sites correspond to instances of observations. In the aligned sequences, each column would correspond to one site. Therefore, this assumption is just the common IID assumption of data points in machine learning. We will clarify these in our revision.
>
> Q3: In the proposed method section, it is hard to tell which part describes the existing work and which part is proposed by the authors.
>
> A3: Most part of the proposed method section is proposed by the authors, except for some necessary background introduction. To summarize (which we also used as subsection titles), we proposed: (1) the interior node embedding for phylogenetic trees based on Dirichlet energy minimization; (2) a linear time two-pass algorithm for Dirichlet energy minimization on phylogenetic trees; (3) theoretical results regarding the tree topology representation power of the obtained interior node features (mainly the identifiability theorem) (4) two examples of applications of GNN-based learnable topologies features for phylogenetic inference (energy based models for tree probability estimation and branch length amortized inference in VBPI).

---

> ### Author Response · Authors · 2022-11-10
> **Response to Reviewer uXSY (Part 2/2)**
>
> Q4: And if I understand correctly, the learning of node features based on the Dirichlet energy does require the knowledge of the phylogenetic tree structure. However, I thought the original problem is to find the appropriate topological structure?
>
> A4: Dirichlet energy minimization is to generate raw node features for phylogenetic trees, which together with GNNs can provide learnable topological features that automatically adapt to downstream tasks.
> Note that Dirichlet energy minimization can be done for any tree topology (without knowing which one is more appropriate), and these learnable topological features could help us to efficiently leverage the topological information of trees to facilitate phylogenetic inference, which is automatically done during the training processes.
> This is somewhat like computing the likelihood function requires the knowledge of the phylogenetic tree structure, but it does not stop us from finding appropriate phylogenetic trees via maximum likelihood estimation.
>
> Furthermore, the phylogenetic tree topology does not necessarily provide appropriate representations for itself or its local topological structures (e.g., edges). In the VBPI example, the branch lengths are associated with the edges on the tree topologies. Parameterizing branch lengths across different tree topologies (i.e., amortized inference) therefore requires finding appropriate representations of the edges across tree topologies which was previously done using heuristic features (e.g., Split and PSP). In this case, learnable topological features allows a more flexible amortization of branch lengths over tree topologies (see the reduced amortization gaps shown in Figure 3) that does not depend on heuristic features.
>
> Q5: I also found the experiment results not convincing enough. First, the author tested multiple GNN algorithms to compare with the results. The authors need to consider the multiple hypothesis testing effect. Second, in table 1, many results are not significant even though it is highlighted in bold. The difference between the proposed method with the original SPLIT and PSP is quite small in most cases.
>
> A5: As mentioned by another reviewer, the GNN variants we tested in our experiments is more as an ablation study and they actually performed similarly and outperformed the heuristic feature baselines, especially in terms of ELBOs. We want to clarify that the improvements of ELBOs for GNNs are indeed significant considering the marginal likelihoods they approach. For the marginal likelihood estimates, all methods provide estimates for the same marginal likelihood, and better approximation would lead to **smaller variance**. Because of this, we agree that marginal likelihood estimates may not be a perfect evaluation metric that reflects the approximation quality well. A more appropriate evaluation metric is the amortization gap, which we illustrated in the middle and right plots in Figure 3. More detailed comparison can be found in appendix D.
>
> We also want to emphasize that our method can automatically learn efficient topological features for downstream tasks, while heuristic feature baseline methods like SPLIT and PSP are based on heuristic features that may require significant design effort and domain expertise.
>
> Q6: Minor comment: There is a type after equation (1). I believe the root node is \tao instead of r?
>
> A6: Thanks for catching this typo! The root node is $r$, which can be any interior node if the tree is unrooted and the substitution model is reversible. We used $\rho$ for the root node in equation (1) (this is the typo), and will modify accordingly in our revision.
>
> Q7: The authors did experiments with several GNN algorithms, but the results look not strong enough (and interestingly GIN does not have much advantage in this case)
>
> A7: We are also curious about why GIN does not have much advantage over other GNN variants in this case. One explanation would be that the local neighborhood structure of phylogenetic trees are similar (interior nodes all have three neighbors in unrooted trees), making the multiset identifiability of GIN less advantageous. Designing more powerful GNNs for phylogenetic models would be an interesting future direction.

---

### Official Review · Reviewer_Qbs2 · 2022-10-25

**Confidence:** 4
**Correctness:** 4
**Technical Novelty And Significance:** 4
**Empirical Novelty And Significance:** 3
**Recommendation:** 8

**Clarity, Quality, Novelty And Reproducibility:**

Clarity: good to excellent

Quality: good to excellent

Originality: good, some results that are "obvious in hindsight" but nevertheless important to state; the adoption of GNNs is novel I believe

Reproducibility: no code has been provided, but a clear enough description is given to allow for the results to be reproduced by a sophisticated practitioner.

**Strength And Weaknesses:**

Strengths:

Rigorous proofs of the main theoretical results (numerical stability of the learned topological features, identifiability of the tree topology from a set of linearly independent features, and the maximum principle) - though many of these are "obvious in hindsight" to anyone familiar with Laplace PDEs, which are worth mentioning as an explicit connection (the matrix equation you are working with being a discrete Laplacian) - also, while I cannot think of a reference off the top of my head, these results are unlikely to all be novel; I derived the first one myself when working with numerical characters (copy numbers) when minimising the squared distance along the edges, but never published the results.

Good exposition overall, very clear explanation of what role every part of the paper plays (but the details of splits and pairwise split pairs would work better if moved into the main text rather than left in the appendix).

Reasonable testing approach which makes it directly comparable with previous work on the topic as well as other state-of-the-art methods.

Weaknesses:

It is not clear how reasonable the assumption of linear independence among the tip vectors would be. If I understand the one-hot encoding  right, having two positions, say x and y, in the alignment where 00, 01, 10 and 11 all occur at least once among the leaves would contradict linear independence. This may be the only possible violation of linear independence; the non-existence of such a pair of positions is a well-known condition, called the "perfect phylogeny" condition. If this is indeed accurate then proving a lemma to that effect would be powerful. As an aside, it is not clear what happens when linear independence fails; does the algorithm break down or can it deal with the situation?

The GNN's are not very well-described, a lot of prior knowledge is taken for granted; I understand that this is a conference paper with a page limit, but a bit more explanation would still be helpful (in particular, what makes EDGE different from other models, or what the aggregation and summarisation functions look like, could help the reader make more sense of the approach).

The optimal results in the table seem to differ only a tiny amount relative to the sub-optimal results; this is worth mentioning explicitly and also speculating about the implications (i.e. sure, the new method seems to be performing better, but only by a tiny amount); maybe this is due to the small tree size. Alternatively, a secondary evaluation metric could be considered in order to showcase the method's advantages.

A number of relevant papers that also carry out machine learning or automated inference of tree topology features (Matsen's 2007 "Optimization over a class of tree shape statistics", and Hayati et al, Royal Society B, 2022 for a more modern example) are not cited - this is not necessarily an issue as they are looking at a different class of problems, but it would be helpful for setting the context to mention that this (broadly speaking) is an active area of research, especially as many of the other references used are somewhat dated.

**Summary Of The Paper:**

The paper provides a novel method for an efficient learning of topological features and then leveraging the power of graph neural networks (GNNs) to conduct tree inference (i.e. simultaneously learning good tree topologies and branch lengths). This approach is shown to perform better than several competitors.

**Summary Of The Review:**

This is a strong paper, both theoretically and experimentally, and with a bit of clean-up to address the comments above it could really shine.

---

> ### Author Response · Authors · 2022-11-10
> **Response to Reviewer Qbs2 (Part 1/2)**
>
> Thank you for your thoughtful review and valuable feedback. Below are the answers to your comments:
>
> Q1: Rigorous proofs of the main theoretical results (numerical stability of the learned topological features, identifiability of the tree topology from a set of linearly independent features, and the maximum principle) - though many of these are "obvious in hindsight" to anyone familiar with Laplace PDEs, which are worth mentioning as an explicit connection (the matrix equation you are working with being a discrete Laplacian) - also, while I cannot think of a reference off the top of my head, these results are unlikely to all be novel; I derived the first one myself when working with numerical characters (copy numbers) when minimising the squared distance along the edges, but never published the results.
>
> A1: Yes, you are absolutely right! The linear system can be viewed as a discrete Laplacian and that is why the Extremum Principle holds. In that sense, it is similar to elliptic PDEs. However, these analyses are only to elucidate the structure of the embedded features which pave the way to our main theoretical result: the identifiability of the tree topology using the embedded features for the interior nodes. This, as far as we are concerned, is new.
>
> Q2: It is not clear how reasonable the assumption of linear independence among the tip vectors would be. If I understand the one-hot encoding right, having two positions, say x and y, in the alignment where 00, 01, 10 and 11 all occur at least once among the leaves would contradict linear independence. This may be the only possible violation of linear independence; the non-existence of such a pair of positions is a well-known condition, called the "perfect phylogeny" condition. If this is indeed accurate then proving a lemma to that effect would be powerful. As an aside, it is not clear what happens when linear independence fails; does the algorithm break down or can it deal with the situation?
>
> A2: Thanks for raising this up! The one-hot encoding is for the taxa, regardless of their specific alignments. For example, if there are four taxa $A, B, C, D$ that represent four species. One can use one-hot encoding $[1,0,0,0],[0,1,0,0],[0,0,1,0],[0,0,0,1]$ for $A, B, C, D$ respectively. Therefore, the tip vectors using one-hot encoding would be naturally linearly independent. We will make this clearer in our revision. If linear independence fails, the two pass algorithm still works but the identifiability of the tree topology from the embedded interior features would fail. The simplest counterexample is the case where all tip vectors are the same. Then the embedded interior node features will be the same vector, regardless of the tree topology.
>
> Q3: The GNN's are not very well-described, a lot of prior knowledge is taken for granted; I understand that this is a conference paper with a page limit, but a bit more explanation would still be helpful (in particular, what makes EDGE different from other models, or what the aggregation and summarisation functions look like, could help the reader make more sense of the approach).
>
> A3: Thanks for the suggestion! We will add more explanation in our revision. Regarding EDGE, what makes it different from other models is that it can generate edge features that describe the relationships between a node and its neighbors, which fits the branch length parameterization task well. We will describe the aggregation and summarization function for different graph convolution operators in the appendix. Note that we treat this work more as an introduction of GNNs to phylogenetic inference, and hence do not dive deeply into specific GNN architectures. Designing GNNs that adapt to phylogenetic models would be an interesting future direction.

---

> ### Author Response · Authors · 2022-11-10
> **Response to Reviewer Qbs2 (Part 2/2)**
>
> Q4: The optimal results in the table seem to differ only a tiny amount relative to the sub-optimal results; this is worth mentioning explicitly and also speculating about the implications (i.e. sure, the new method seems to be performing better, but only by a tiny amount); maybe this is due to the small tree size. Alternatively, a secondary evaluation metric could be considered in order to showcase the method's advantages.
>
> A4: We want to clarify that the ELBOs for GNNs are indeed significantly improved compared to the heuristic feature baselines (Split and PSP) considering the marginal likelihoods they approach. For the marginal likelihood estimates, all methods provide estimates for the same marginal likelihood, and better approximation would lead to **smaller variance**. We agree that marginal likelihood estimates may not be a perfect evaluation metric that reflects the approximation quality well. The main benefit of GNN-based learnable topological features for VBPI is for amortized inference of the branch lengths over tree topologies. The middle and right plots in Figure 3 demonstrate that our method significantly reduced the amortization gaps compared to heuristic feature baselines (Split and PSP). More detailed comparison can be found in appendix D.
>
> We also want to emphasize that our method does not depend on heuristic features that may require significant design effort and domain expertise.
>
> Q5: A number of relevant papers that also carry out machine learning or automated inference of tree topology features (Matsen's 2007 "Optimization over a class of tree shape statistics", and Hayati et al, Royal Society B, 2022 for a more modern example) are not cited - this is not necessarily an issue as they are looking at a different class of problems, but it would be helpful for setting the context to mention that this (broadly speaking) is an active area of research, especially as many of the other references used are somewhat dated.
>
> A5: Thanks for introducing these related works on learning tree topology features. We will add them to the introduction in our revision.
>
> Q6: Reproducibility: no code has been provided, but a clear enough description is given to allow for the results to be reproduced by a sophisticated practitioner.
>
> A6: Thanks for the appreciation of the clarity of our description. The code is also provided in the supplementary material, please check it out :)

---

### Official Review · Reviewer_b2oj · 2022-11-04

**Confidence:** 3
**Correctness:** 4
**Technical Novelty And Significance:** 3
**Empirical Novelty And Significance:** 3
**Recommendation:** 8

**Clarity, Quality, Novelty And Reproducibility:**

***Clarity*** The paper is fairly written, without any major clarification issues. Both the theoretical and the empirical parts are easy to follow.

***Novelty*** The problem of phylogenetic inference is a very important problem, as it can help towards better understanding  of the evolutionary history of protein sequences, and genes. Although deep learning has been used before for tasks on phylogenetic trees, the utilization of graph-based machine learning methods is, to my knowledge, yet unexplored. The present paper suggests a novel approach based on efficiently computed raw node features and a graph neural network architecture to exploit the bias of the tree topology.

***Quality*** The paper presents both theoretical and empirical validation. The theoretical part solidifies the use of the computed features based on the Dirichlet energy minimization method for the characterization of phylogenetic tree topologies. In the empirical evaluation, the authors, show the effectiveness of the combination of graph-based and energy-based methods for tree topologies.

***Reproducibility*** The code and data are attached in the supplementary material. They are clean, making a rerun of the experiments feasible.

**Strength And Weaknesses:**

***Strengths***
1. The authors provide an efficient algorithm (based on the Thomas Algorithm [Thomas, 1949]) for computing raw node features of the initial phylogenetic tree nodes, based on the Dirichlet energy minimization objective. This algorithm gives a linear-time complexity for the pre-computation of the graph input of the phylogenetic inference.
2. The authors suggest the utilization of graph neural networks for learning structural representations of the intermediate states in a tree inference task. They show theoretically that the input raw node features yield representations that can discriminate different tree topologies.
3. Given the learned structural representations, the authors propose two successful ways of their utilization: 1) for the tree probability estimation through an energy-based model, that takes as input the learned node representations, 2)  for the branch length parameterization in the bayesian task, by providing the output GNN representations as statistical measures.

***Weaknesses***
1. The phase of feature initialization (through the linear two-time pass algorithm) is independent of the later graph-based representation learning method. What would be the impact of an initialization method that is dependent on the downstream task?
2. The experimental evaluation both for the probability estimation and the VBPI tasks is performed more as an ablation study over various graph neural network encoders. To my knowledge, there are some deep learning approaches that deal with the problem of phylogenetic inference [Solis-Lemus 2022, Jiang 2022, Fioravanti 2017]. Could the authors discuss how their method is compared with theirs or explain why these alternative approaches may not be applicable?

- [Thomas 1949] Elliptic problems in linear difference equations over a network. 1949
- [Solis-Lemus 2022] Accurate Phylogenetic Inference with a Symmetry-preserving Neural Network Model. 2022
- [Jiang 2022] Learning Hyperbolic Embedding for Phylogenetic Tree Placement and Updates. 2022
- [Fioravanti 2017] Phylogenetic convolutional neural networks in metagenomics. 2017

**Summary Of The Paper:**

***Paper Summary*** The authors suggest a structural representation model for the task of phylogenetic inference by the utilization of graph neural networks architectures. For the initial node attributes, they propose the construction of raw features based on the Dirichlet energy minimization. Using message passing steps, their model is able to learn structural features of the phylogenetic trees, depending on the downstream task. In an empirical level, the authors evaluate their method in synthetic and real data, highlighting a superior performance.

***Contribution*** The paper has two main contributing components on the problem of phylogenetic inference. 1) The efficient computation of theoretically expressive tree node features, based on the minimization of the Dirichlet energy. The authors prove that using the computed Dirichlet energy-based features can provide representations, capable of distinguishing different tree topologies (Theorem 2). 2) The utilization of a graph neural network architecture for learning interior node representations, that can later be used in the downstream task.

**Summary Of The Review:**

The paper presents a novel methodology for performing inference on phylogenetic trees. Their main contributions are twofold: they provide an efficient method for computing theoretically expressive initial node features on the phylogenetic trees and they propose a graph neural network-based model for learning representations, that are later used in downstream tasks. Both the theoretical and empirical evaluation seem solid, while the novelty and importance of the method is fair. That being said, I recommend the acceptance of the paper to the venue.

---

> ### Author Response · Authors · 2022-11-10
> **Response to Reviewer b2oj**
>
> Thank you for your thoughtful review and valuable feedback. We address your specific questions and comments below
>
> Q1. The phase of feature initialization (through the linear two-time pass algorithm) is independent of the later graph-based representation learning method.
> What would be the impact of an initialization method that is dependent on the downstream task?
>
> A1. Thanks for this nice question! Yes, currently the initialization is independent and adaption to downstream tasks purely relies on GNNs.
> It would be beneficial for specific tasks if some task information can be integrated into the initialization (e.g. through the weights on different edges for Dirichlet energy minimization).
> However, this may also make it less generalizable across different tasks.
>
> Q2. The experimental evaluation both for the probability estimation and the VBPI tasks is performed more as an ablation study over various graph neural network encoders.
> To my knowledge, there are some deep learning approaches that deal with the problem of phylogenetic inference [Solis-Lemus 2022, Jiang 2022, Fioravanti 2017].
> Could the authors discuss how their method is compared with theirs or explain why these alternative approaches may not be applicable?
>
> A2. Thanks for providing these related deep learning methods for phylogenetic inference!
>
> (1) The method proposed by Solis-Lemus et.al 2022 uses symmetry-preserving neural networks to explicitly encode permutation invariance of phylogenetic trees which is also the core idea behind GNNs (the message passing step is permutation invariant of the neighboring nodes). However, it is mainly designed for 4-taxa datasets and is challenging to extend to n-taxa datasets as pointed out by the authors.
>
> (2) The method proposed by Jiang et.al 2022 learns hyperbolic embeddings for the leaf nodes and is mainly designed for phylogenetic tree placement and update. The resulting embeddings for the leaf nodes would change if the backbone tree topology is different. Our method is for learning topological features of phylogenetic trees (either the entire graph or local structures like edges) that generalize across tree topologies. Therefore, the embeddings for the leaf nodes are fixed (i.e., one hot encoding) in our case.
>
> (3) The method proposed by Fioravanti 2017 uses CNNs that incorporate distance information from the phylogenetic tree for the classification of metagenomics data. The method assumes the phylogenetic tree is known and fixed. The phylogenetic information then is passed to CNNs via the induced distance matrix. Our method is to learn tree topology representations for phylogenetic inference (i.e., learning the phylogenetic tree), and hence requires generalization across tree topologies.
>
> We will add those references in our revision.

---

### Author Response · Authors · 2022-11-15
**Revision summary**

We thank all reviewers for the constructive feedback. We have revised the paper (the modified parts are highlighted in blue), and have incorporated their suggestions with the following major changes:

- We reorganized the Background section and used a separated notation paragraph to clarify concepts and notations for phylogenetic models.

- We added discussions on related papers on learning tree topology features to the introduction and deep learning approaches for phylogenetic inference to Appendix E.

- We added more detailed descriptions of the aggregration and summarization functions for different graph convolutional operators to Appendix F.

- We clarified a bit why EDGE was used as an example of GNN variants for branch length parameterization in VBPI.


We hope our revision has adequately addressed the reviewers' questions and concerns, and look forward to reading any further comments.

---

### Decision · Program_Chairs · 2023-01-20

**Decision:**

Accept: poster

**Justification For Why Not Higher Score:**

See the above-mentioned weaknesses.

**Justification For Why Not Lower Score:**

See the above-mentioned strengths.

**Metareview: Summary, Strengths And Weaknesses:**

This paper adopts GNNs to help learn the structure information of phylogenetic trees and perform inference on downstream tasks. For initial node features, the approach transforms raw input features using the Dirichlet energy minimization. The model uses MPNN layers to learn the structural features of the phylogenetic trees, depending on the downstream task.

**Strengths:**

* Reviewers appreciate that authors showed theoretically that the input raw node features yield representations that can discriminate different tree topologies. The efficient computation of theoretically expressive tree node features is based on the minimization of the Dirichlet energy. The authors prove that using the computed Dirichlet energy-based features can provide representations capable of distinguishing different tree topologies.
* The MPNN is used for learning interior node representations that can later be used in the downstream tree inference tasks.
* Reviewers agree that the paper is well-written overall and nicely organized with a very clear explanation of what role every part of the paper plays.

**Weaknesses:**
* The optimal results in the table seem to differ only a tiny amount relative to the sub-optimal results. The new method seems to perform better, but only by a tiny amount, which might be due to the small tree size.  Several reviewers pointed out that experimental results are not particularly strong, however, the method also has other advantages over heuristic methods.
* Secondary evaluation metrics could be considered in order to better showcase the method's advantages.

**Note From Pc:**

if the above contains the word "oral" or "spotlight" please see: "oral" presentation means -> notable-top-5% and "spotlight" means -> notable-top-25%. As stated in our emails, we are disassociating presentation type from AC recommendations